# Chronic Thromboembolic Pulmonary Hypertension: A Review of the Multifaceted Pathobiology

**DOI:** 10.3390/biomedicines12010046

**Published:** 2023-12-24

**Authors:** Hakim Ghani, Joanna Pepke-Zaba

**Affiliations:** Pulmonary Vascular Disease Unit, Royal Papworth Hospital, Cambridge CB2 0AY, UK; joanna.pepke-zaba@nhs.net

**Keywords:** chronic thromboembolic pulmonary hypertension, pathobiology, genetics, inflammation, defective angiogenesis

## Abstract

Chronic thromboembolic pulmonary disease results from the incomplete resolution of thrombi, leading to fibrotic obstructions. These vascular obstructions and additional microvasculopathy may lead to chronic thromboembolic pulmonary hypertension (CTEPH) with increased pulmonary arterial pressure and pulmonary vascular resistance, which, if left untreated, can lead to right heart failure and death. The pathobiology of CTEPH has been challenging to unravel due to its rarity, possible interference of results with anticoagulation, difficulty in selecting the most relevant study time point in relation to presentation with acute pulmonary embolism (PE), and lack of animal models. In this article, we review the most relevant multifaceted cross-talking pathogenic mechanisms and advances in understanding the pathobiology in CTEPH, as well as its challenges and future direction. There appears to be a genetic background affecting the relevant pathological pathways. This includes genetic associations with dysfibrinogenemia resulting in fibrinolysis resistance, defective angiogenesis affecting thrombus resolution, and inflammatory mediators driving chronic inflammation in CTEPH. However, these are not necessarily specific to CTEPH and some of the pathways are also described in acute PE or deep vein thrombosis. In addition, there is a complex interplay between angiogenic and inflammatory mediators driving thrombus non-resolution, endothelial dysfunction, and vascular remodeling. Furthermore, there are data to suggest that infection, the microbiome, circulating microparticles, and the plasma metabolome are contributing to the pathobiology of CTEPH.

## 1. Introduction

Chronic thromboembolic pulmonary disease (CTEPD) is characterized by chronic organized thromboembolic material in the pulmonary arterial tree [1,2,3,4]. The non-resolving, organized thrombi resulting in pulmonary artery (PA) obstructions and additional microvascular remodeling may lead to the development of pulmonary hypertension, and this condition is referred to as chronic thromboembolic pulmonary hypertension (CTEPH) [1,2,3,4,5,6]. Risk factors for CTEPH have been well documented; however, how they biologically influence the development of chronic thrombi are poorly understood [1,3,4,7,8]. About 75% of patients with CTEPH have a history of acute pulmonary embolism (PE) and although residual thrombi or perfusion defects after acute PE are not uncommon, the relationship between acute PE and progression to chronic thrombi and CTEPH is difficult to elucidate [3,8,9,10,11]. CTEPH patients without a history of acute PE or deep vein thrombosis (DVT) may have a subclinical thromboembolic event. The incidence of CTEPH after acute PE is between 0.1 to 9.1%, but the exact quantification of incidence is a complex task due to both underdiagnosis and in some cases, overestimation [1,12,13]. The most recent meta-analysis showed that the pooled CTEPH incidence in survivors of acute PE was 2.7% [14]. The question as to why only a minority of post-acute PE patients develop CTEPH is still poorly understood and there is no concrete data showing whether the natural history from acute PE to CTEPD/CTEPH involves a single or multiple independent pathways.

Large registries have reported associated medical conditions as risk factors for CTEPH development. This includes malignancy, chronic inflammatory disease, low-grade infections of pacing wires and ventriculo-atrial shunts, antiphospholipid antibodies, splenectomy, thrombophilic disorder, and non-O blood group [8,15,16]. These associated clinical risk factors seen in CTEPH overlap with acute PE and venous thromboembolism (VTE). Although helpful in finding predisposing factors for CTEPH in clinical practice, they do not explain the biological transition from acute PE to chronic thrombi and CTEPH. 

The pathobiological pathways leading to CTEPH have been challenging to elucidate as it is a rare condition and pathogenic mechanisms are usually studied in patients with established CTEPH. Furthermore, selection of the most relevant time points to study the natural history from acute PE to CTEPD/CTEPH are unknown. Additionally, patients are treated with anticoagulation therapy after acute PE which interferes with coagulation pathways. Although there are some animal models attempting to replicate specific pathophysiological and possible pathobiological pathways contributing to CTEPH development including vascular remodeling, there are no animal CTEPH models that encompass the natural pathobiological process in its entirety [17,18,19]. Genetic studies, particularly genome-wide association studies, may provide the most accurate identification of pathobiological pathways in CTEPH.

Numerous pathobiological mechanisms have been proposed in the published medical literature which may be contrasting. Furthermore, many proposed pathological pathways may overlap with acute PE and VTE which therefore does not explain the unique mechanisms in CTEPH development. In addition, some studies attempting to describe the pathobiology in CTEPH do not compare cohorts of patients which can share similar pathological pathways including acute PE/VTE and pulmonary arterial hypertension (PAH). Due to the complex nature of the development of CTEPH, multifaceted pathological pathways have been explored to comprehend the biological process. In this article, we review the most relevant research findings in CTEPH, make the vital comparison with diseases that can share similar pathological pathways resulting in challenges with interpreting proposed pathological mechanisms, and discuss the advances in understanding the pathobiology in CTEPH. 

## 2. Pathophysiology and Histopathology

The pulmonary artery fibrotic obstructions can cause an increase in pulmonary arterial pressure and subsequent vascular remodeling may lead to microvasculopathy (Figure 1) [2,4]. Microvasculopathy with remodeling of the muscular pre-capillary pulmonary arteries can occur due to exposure to high pressure from pulmonary flow redistribution to unobstructed pulmonary arteries and can resemble changes seen in pulmonary arterial hypertension (PAH). Microvasculopathy distal to obstructed pulmonary arteries can be explained by anastomoses from systemic bronchial arteries to the pulmonary circulation distal to the obstruction [3,4,6,20]. Untreated, the pulmonary artery obstructions and microvasculopathy can lead to a rise in pulmonary arterial pressure and pulmonary vascular resistance, and finally, to right heart failure and death [4].

Historically, CTEPH is described as the ‘two-compartment model’ with chronic clots occluding the pulmonary arteries and small vessel microvasculopathy [6,21]. However, CTEPH currently may be better described as ‘tri-compartmental’ from a management perspective (Figure 1). The proximal fibrotic thromboembolic component can be approached via pulmonary endarterectomy (PEA), the distal subsegmental occlusions which are not surgically accessible but are targets for balloon angioplasty (BPA) and the microvascular component for pulmonary hypertension medical therapy [22]. 

Histological examinations of pulmonary endarterectomy specimens have revealed the complex structures of chronic thrombi and remodeling of the pulmonary vasculature. Chronic thrombi in larger elastic pulmonary arteries result in obstructive hypertrophic remodeling, and neovascularization of the thrombus from the systemic vasa vasorum is also described [2,4,20,21]. Organized fibrotic thrombi which adhere tightly to the pulmonary artery consist primarily of collagen, elastin, and inflammatory cells, and differ from acute PE which is largely erythrocytes and platelets in a fibrin mesh [2,3,4]. Histology of distal and microvascular compartments show pathological remodeling with eccentric intimal fibrosis, colander-like lesions, and pulmonary capillary hemangiomatosis-like remodeling [2,4,5,20]. Plexiform lesions, which are typically seen in pulmonary arterial hypertension, have also been found by some [2,4,20]. Interestingly, histological studies show differential distribution in inflammatory cells where macrophages, T-lymphocytes, and neutrophils accumulation are mainly found in atherosclerotic and thrombotic lesions, and B cells are only found in thrombotic lesions, whereas recanalized and intimal lesions are almost devoid of inflammatory cells [23].

## 3. Role of Fibrinogen, Thrombosis, and Fibrinolysis in Thrombus Organization 

Variation in fibrinogen morphology resulting in resistance to fibrinolysis may offer a partial explanation for the pathological transition from acute PE into chronic thrombi and CTEPH. Five fibrinogen variants with corresponding heterozygous gene mutations have been discovered in CTEPH compared to healthy controls: *Bβ P235L/γ R375W, Bβ P235L/γ Y114H, Bβ P235L, Aα L69H*, and *Aα R554H*, resulting in a disorganized fibrin structure and fibrinolytic resistance [24,25]. Although these fibrinogen variants have not been found to be associated with VTE in other studies, it is unclear whether they are specific to CTEPH and would require comparison with a VTE cohort. Morris et al. has shown alterations to the fibrin structure, with persistence of the N-terminus of the β-chain of fibrinogen in CTEPH, which may result in the relative resistance to plasmin-mediated fibrinolysis due to inaccessibility of plasmin cleavage sites [26]. N-terminus persistence of the β-chain of fibrin has been implicated in cell signaling, angiogenesis, platelet spreading, and the growth of fibrin polymers. Furthermore, the interaction of fibrinogen, fibrin, and thrombin with pulmonary artery endothelial cells and smooth muscle cells regulate intracellular calcium, which controls cell migration, proliferation, and contraction, and may contribute to vascular changes observed in CTEPH patients [27,28].

The non-resolution of chronic thrombi in CTEPH would instinctively bring about the discussion of hypercoagulability in the thrombosis and fibrinolysis pathways. However, classic risks of thrombophilia with mutations of antithrombin, protein C, protein S, prothrombin, and factor V are not independently associated with CTEPH [6,8,21,29]. A higher frequency of lupus anticoagulant and antiphospholipid antibodies, which increase the risk of thromboembolism, is present in 10–20% of CTEPH patients but does not fully explain the biologic process of the development of chronic thrombi [29]. Elevated levels of coagulation factor VIII have been found in CTEPH but have also been associated with pulmonary arterial hypertension (PAH) [8,30,31]. Of significance, higher levels of von Willebrand factor (vWF), which activates factor VIII and is a mediator of platelet adhesion and aggregation, even after PEA, may suggest its more central role in the development of chronic thrombi [30,31,32,33]. 

Platelets play a crucial part in the prothrombotic milieu in CTEPH by being chronically activated with a higher turnover, exhibiting hyper-responsiveness to thrombin stimulation and increased spontaneous platelet aggregation [31,34,35]. Platelets, in addition, may play an inflammatory mediatory role in CTEPH by inducing vascular cell adhesion molecule-1 (VCAM-1) expression in endothelial cells [35]. 

The systemic fibrinolysis pathway is typically considered not to be affected in CTEPH, but imbalances in local expression of tissue-type plasminogen activator (tPA) and type 1 plasminogen activator inhibitor (PAI-1) have been observed [36,37]. Elevated PAI-1 expression within PEA specimens may suggest its inhibitory role in thrombolysis failure through the stabilization of chronic thrombi [38]. A study has shown elevated levels of plasma thrombin-activatable fibrinolysis inhibitor (TAFI) in CTEPH compared to controls [39]. The activated form of TAFI (TAFIa) removes the fibrin C-terminal lysine binding sites for tPA and plasminogen, decreasing the ability of fibrin to stimulate fibrinolysis. Yaoita et al. also showed that clot lysis time improves with CPI-2KR, an inhibitor of TAFIa [39].

## 4. Metabolomics- Angiogenesis and Inflammation

Chronic inflammatory conditions increase the risk of CTEPH; there is salient evidence of inflammatory bowel disease as a predisposing risk factor [8,16,40]. Acute and chronic inflammatory conditions result in a vicious cycle of hemostasis, innate immunity, platelet activation, endothelial activation, and coagulation, giving a higher risk for acute and recurrent VTE [41,42]. However, as to why the aberrant immuno-thrombosis and thromboinflammation in a relatively small number of patients with chronic inflammation leads to the development of CTEPH remains to be elucidated. There appears to be intricate interplay between inflammatory and angiogenic mediators in CTEPH, which can conceptually narrate inflammatory thrombosis and defective angiogenesis as the etiopathology [43,44,45,46]. 

A mice model of stagnant flow venous thrombosis with inferior vena cava (IVC) ligation showed deficient thrombi angiogenesis and delayed thrombi resolution with endothelial cell-specific deletion of vascular endothelial growth factor receptor 2/kinase insert domain protein receptor (VEGF-R2/Kdr) [47]. In the same study, Alias et al. showed that white CTEPH thrombi from PEA had lower Kdr gene expression. Another mice model with stagnant venous flow showed deficient platelet endothelial cell adhesion molecule-1 (PECAM-1), resulting in larger thrombi and misguided resolution [48]. PECAM-1 is involved in leukocyte migration, inflammation, and angiogenesis, and its expression is reduced in CTEPH patients [47,48,49]. In addition, angiostatic factors are found in PEA specimens, including platelet factor 4 (PF4), collagen type I, and interferon-gamma-inducible 10 kD protein (IP-10), as well as increased levels of CXCR3, the receptor for PF4 and IP-10 [50]. PF4, collagen type I, and IP-10 cause altered pulmonary artery endothelial cells calcium homeostasis and affect its proliferation, migration, and vessel formation, therefore suggesting endothelial dysfunction and consequentially, decreased angiogenesis [50].

Recently, Hadinnapola et al. found significant serum elevation of angiopoietin (Ang) 2, interleukin (IL) 8, and tumor necrosis factor (TNF) α in CTEPH patients compared to a control group [43]. Increased Ang2 expression, an antagonist ligand of the endothelial-specific TIE2 receptor, released from endothelial Weibel–Palade bodies upon various stimuli including hypoxia, thrombin, and inflammation, may support the evolution of acute PE to CTEPH by delaying venous thrombus resolution and contributing to thrombofibrosis [43,51,52]. Furthermore, Ang2 acts as an endothelial autocrine inflammatory regulator by sensitizing endothelial cells toward elevated levels of TNFα and modulating TNFα-induced expression of endothelial adhesion molecules [43,53,54].

Neutrophil extracellular traps (NETs) formation, NETosis, which is proinflammatory, prothrombotic, and proangiogenic, is elevated in both plasma and PEA specimens in CTEPH [55,56,57]. Similar findings were also found in idiopathic PAH (IPAH) patients. NETosis resulting from at least three possible mechanisms, cell lytic, vesicle-mediated, and mitochondrial release, of neutrophils result in a net of chromatin fibers with neutrophil secretory granules containing elastase and myeloperoxidase which can induce nuclear factor (NF)-κB and transforming growth factor (TGF) β in CTEPH [55,56,58]. Interestingly, a small study showed that NETs form predominantly in the organizing stage of thrombi development, which require further research as a possible important mechanism in CTEPH [59]. Although NETs support chronic thrombi development through thrombofibrosis and promoting thromboinflammation, NETosis seems to be proangiogenic which is counterintuitive to the defective angiogenesis narrative in CTEPH [55,56]. However, this may explain the variability of revascularization seen in chronic thrombi. Degradation of NETs with DNases may prevent chronic clot formation when used in the acute phase [55,56,58].

Elevated high-sensitive CRP (hs-CRP), although not specific to CTEPH, is more than a bystander marker of inflammation. In CTEPH, elevated levels of hs-CRP are associated with increases in pulmonary smooth muscle cell proliferation and pulmonary endothelial cell adhesion by intracellular adhesion molecule-1 (ICAM-1) and secretion of endothelin-1 and vWF [43,60]. Furthermore, lectin-like oxidized low-density lipoprotein receptor (LOX)-1, a hs-CRP ligand, is overexpressed in CTEPH and can contribute to endothelial dysfunction through activation of proinflammatory genes [60,61]. Additionally, hs-CRP also induces pulmonary endothelial cell activation of NF-κB, which has been found to be a regulator of inflammatory mediators in CTEPH [45,62,63,64].

There is an obvious proinflammatory state in CTEPH, as described above, and significant upregulation in serum inflammatory mediators including IL-6, IL-8, IP-10, monokine induced by interferon-γ (MIG), macrophage inflammatory protein (MIP) 1α, and matrix metalloproteinase (MMP) 9 [23,65,66]. However, we can question whether this is independently contributing to the etiopathology of developing chronic thrombi or whether it is due to right ventricular failure, as heart failure is known to be associated with inflammation and elevated inflammatory mediators [67,68,69,70]. As PEA specimens also show higher levels of cytokines including IL-6, IL-8, IL-1β, MCP-1, IP-10, MIP1α, and RANTES, we can assume an independently occurring inflammatory process in CTEPH [45,65,66]. 

Interestingly, angiogenic factors and inflammatory mediators can also act as possible biomarkers to determine residual PH post-PEA. Quarck et al. found significantly lower VEGF in CTEPH patients with residual pulmonary hypertension (PH) post-PEA, and VEGF was inversely correlated with mean pulmonary artery pressure (mPAP) 3 days after PEA [23]. However, this may be influenced by the post-operative inflammatory state. Of particular significance, pre–PEA Ang2 corresponds with pre-PEA mPAP and pulmonary vascular resistance (PVR) and is associated with residual PH post-PEA [43]. With possible important translational clinical implications, this needs more space in prospective CTEPH research.

## 5. Microbiome and Infection

Ventriculo-atrial shunts and pacemaker-led infections are risk factors for CTEPH, where the low-grade infection leading to chronic inflammation could partially explain the etiopathology of chronic thrombi formation [16,71]. Bonderman et al. detected staphylococcal DNA in six out of seven thromboemboli specimens harvested from PEA in CTEPH patients with ventriculo-atrial shunts [72]. In the same study, a mouse model of stagnant flow venous thrombosis with IVC ligation showed that staphylococcal infection delayed thrombus resolution and the upregulation of TGF-β and connective tissue growth factor (CTGF) [72]. In addition, NETs appear to play a role in thrombofibrosis, shown by TGF-B overactivity in an IVC ligation mouse model with staphylococcal infection, resulting in larger thrombi and less resolution [56]. TGF-β associated with CTGF expression is linked with a multitude of biological processes including cell proliferation, fibrosis, extracellular matrix deposition, and angiogenesis [73,74]. Staphylococci can potentiate inflammation through peptidoglycan-induced release of cytokines by macrophages or alternatively through protein A surface protein mimicking TNF-α [75,76].

The gut microbiome and dysbiosis is known to affect a plethora of medical conditions including cardiovascular diseases and PAH [77,78,79,80]. Plasma endotoxin levels are significantly increased in CTEPH patients which correlates with TNF-α, IL-6, IL-8, and MIP-1α levels [81]. Increased permeability of the intestine, resulting in elevated endotoxin, can promote inflammation by activating macrophages and endothelial cells through NF-κB [79,81]. Bacterial alpha-diversity, which could offer a protective homeostasis role, is reduced in CTEPH patients and may explain the observed elevated plasma endotoxin and cytokine levels [81]. However, whether the altered gut microbiome is the cause or result of CTEPH is unclear. Furthermore, as heart failure and congestion are associated with gut dysbiosis, it remains to be elucidated whether the altered gut microbiome seen in CTEPH is the result of right heart failure and thus requires more research [80]. 

## 6. Microparticle and Plasma Metabolome

Splenectomy is associated with an increased risk of developing thromboembolism and CTEPH, but the transient reactive thrombocytosis seen post-splenectomy, resolving in several weeks, does not explain the risk of thromboembolism [8,82,83,84,85]. It has been hypothesized that abnormal erythrocytes, normally filtered by the spleen, facilitate coagulation and increase the risk of thrombosis through abnormal exposure to phosphatidylserine, an anionic phospholipid of erythrocyte membrane [83,86]. Frey et al. showed that PEA specimens from CTEPH patients with previous splenectomy were enriched with anionic phospholipids such as phosphatidylserine [87]. A post-splenectomy mice model in the same study showed elevated prothrombotic circulating platelet microparticles and leukocyte–platelet aggregates, which may explain the increase in thrombus formation [87]. Delayed thrombi resolution post-splenectomy with the observed defective angiogenesis may be due to phosphatidylserine inhibition of endothelial cells’ DNA synthesis rates and endothelial sprout formation [87]. Although there is an associated risk of CTEPH with splenectomy, not all splenectomy patients develop CTEPH. Therefore, there is likely additional pathogenic processes to circulating microparticles in splenectomy patients who develop CTEPH.

There is an aberrant plasma metabolic profile in CTEPH with altered lipid metabolism [88]. Heresi et al. found similar plasma metabolome dysregulation in CTEPH and IPAH, with increased acyl carnitines, beta-hydroxybutyrate, amino sugars, modified amino acids, and nucleosides. However, CTEPH has higher fatty acids and glycerol, but lower acyl cholines and lysophospholipids, than IPAH [88]. Similarly, Swietlik et al. showed metabolic dysregulation in CTEPH which was mostly observed in IPAH. This included increased modified nucleosides, TCA cycle intermediates, monohydroxy fatty acids, tryptophan, polyamine and arginine metabolites, and decreased sphingomyelin, phosphocholines, and steroid metabolites [89]. Some of these metabolites exhibited plasma gradients from the superior vena cava (SVC) to PA. Only five plasma metabolites were distinct in CTEPH compared to IPAH and CTEPD without PH: 5-methylthioadenosine, N1-methyladenosine, N1-methylinosine, 7-methylguanine, and N-formylmethionine [89]. Metabolites distinguishing CTEPH from healthy and disease controls showed significant associations with clinical measures of disease severity, with the strongest associations of metabolites with SVC to PA gradients [89]. It may be possible that metabolic dysregulation is a consequence of CTEPH resulting from pulmonary hypertension hemodynamics and right heart failure. However, lipid signaling plays a crucial role in thrombosis, inflammation, and tissue remodeling, and studies on heart failure have shown that metabolic feedback can promote disease progression [90,91]. Thus, the metabolic dysregulation in CTEPH, especially abnormal lipid metabolism, would require further investigation to determine its pathobiological role.

## 7. Genetic Background

It is difficult to establish the genetic predisposition of CTEPH as it is not a Mendelian disease. A genome-wide association study (GWAS) is likely the most appropriate way of describing an unbiased pathobiology in CTEPH. However, this would require a large sample size, which is challenging in the context of a rare disease. 

There are studies describing the genetic predisposition of prothrombosis and fibrinolysis resistance in CTEPH, but this may not be specific as the pathogenic mechanism may overlap with VTE and acute PE. As mentioned above, dysfibrogenemia in CTEPH and corresponding heterozygous gene mutations which point toward prothrombosis and fibrinolysis resistance have been shown but would require comparison with VTE [24,92,93]. Although more research is required, single-nucleotide polymorphisms (SNPs) of the ADAMTS13 gene may account for the lower plasma levels of ADAMTS13 and elevated vWF, which persists even after PEA in CTEPH [30,32]. ADAMTS13 regulates vWF’s coagulative role by cleaving the more active ultra-large vWF multimers; therefore, lower plasma levels of ADAMTS13 are associated with thrombotic pathologies [94,95,96]. The inflammation–coagulation axis in CTEPH is supported by evidence of epigenetic regulation of the vWF promoter in CTEPH endothelium. Reduced histone tri-methylation and increased histone acetylation of the vWF promoter facilitate the binding of NF-κB2 and drive vWF transcription, which in turn increase platelet adhesion [33]. This could suggest the epigenetic role in linking inflammation and coagulation pathways to create an environment favoring in situ thrombosis. Although this study compared CTEPH with healthy controls and IPAH, it lacked an acute PE/VTE cohort which could give insight on whether the epigenetic regulation of vWF is unique to the pathogenesis in CTEPH. In addition, the genetic role associating coagulation and inflammation is further supported by elevated tissue factor (TF) gene expression by monocytes in CTEPH which is vital in thrombosis, hemostasis, and endothelial cell signaling [97,98].

The non-O blood group is more common in CTEPH and associated with higher risk of VTE [8,16,99]. The ABO blood group influences vWF biology and therefore the risk for thrombosis, where levels of plasma vWF are lower in blood group O and blood group O vWF has increased susceptibility to ADAMTS13 proteolysis [100]. The thrombosis risk between vWF and blood group seem to be partially mediated through coagulation factor VIII [101]. The *ABO* gene located in relative proximity to the ADAMTS13 gene gives rise to the possibility of its influence on the ADAMTS13–vWF axis. However, there are no differences in ADAMTS13 antigen levels between ABO groups [32]. Although the vWF level was higher in some non-O blood groups in CTEPH, blood group O still had significantly higher vWF than healthy controls [32]. The over-representation of the non-O blood group in CTEPH is not yet explained, but the *ABO* locus is pleiotropic and could influence CTEPH in alternative pathological pathways which are yet to be elucidated. Perhaps chromosome conformation analysis in CTEPH could describe the qualitative and quantitative epigenetic influence of loci interactions in the three-dimensional space of the genome [102].

The role of *bone morphogenetic protein type II receptor (BMPR2)* gene mutation, now well known in hereditary PAH, is controversial in CTEPH and unconfirmed. There are some indications of BMPR2 polymorphism in CTEPH, but they are not corroborated in different studies and are of uncertain significance [103,104,105]. Another reported genetic variant in CTEPH is *angiotensin-converting enzyme (ACE)*, which plays a key role in the renin–angiotensin system and can cause endothelial dysfunction, inflammation, and vascular remodeling [106]. *ACE* deletion allele carrier, resulting in increased circulating and cellular concentrations of ACE, results in a lower 6-min walk distance test result and poorer prognosis in CTEPH, suggesting its possible pathobiological capacity [107].

Studying RNA, and therefore gene expression and regulation, can lead to better molecular mechanism mapping in CTEPH pathogenesis. CTEPH pulmonary artery smooth muscle cells’ (PASMCs) microRNA expression profile using microarray, gene ontology, and pathway analysis show 18 differentially expressed microRNA, 12 upregulated, and 6 downregulated [108]. Of significance is the downregulation of let-7d microRNA, which may repress the proliferation of PASMCs by upregulating p21, arresting the cell cycle at the G1 phase, and therefore possibly affecting vascular remodeling in CTEPH [108]. Eleven cell types are possibly affected in CTEPH with differentially expressed microRNAs and circular RNAs [109]. Miao et al. found that tumor protein p53, ICAM1, amyloid-β precursor protein (APP), integrin subunit β2 (ITGB2), and zyxin have the highest degree of connectivity in the protein–protein interaction network of different cell types in CTEPH [109]. This shows that CTEPH involves multipathogenic pathways; however, which one is predominant, how they interact, and how they differentially affect individual patients is not entirely clear. 

To understand the wider pathological genetic picture in CTEPH, analysis of a larger genetic profile is required. Gene expression profiles of pulmonary artery endothelial cells in CTEPH using oligonucleotide microarrays show 880 upregulated genes and 734 downregulated genes compared to healthy controls [110]. Although multiple genes may be up or downregulated in a disease, its relevance in a pathological mechanism can be difficult to ascertain. Bioinformatics analyses to annotate gene function using gene ontology and pathway analyses to understand the association of genes by drawing signal transduction networks in CTEPH show a significant interaction between Janus kinase 3 (JAK3), guanine nucleotide binding protein alpha 15 (GNA15), mitogen-activated protein kinase 13 (MAPK13), arrestin beta 2 (ARRB2), coagulation factor II receptor (F2R), and VCAM-1 [110]. Therefore, the differential gene regulation in CTEPH, with interplay in their downstream function, can affect inflammation, cell proliferation and migration, and vascular remodeling pathways. However, the limiting factor was not comparing CTEPH in order to distinguish gene regulation with acute PE, VTE, right heart failure, and PAH as conditions with cross-talking pathobiological pathways.

Genetic studies suggest that CTEPH, at least partially, is a heritable multigenetic disease. Recently, results of the first GWAS in CTEPH conducted in a large multinational cohort was published on a pre-print server [111]. *FGG* gene coding for the fibrinogen gamma chain and the *ABO* gene, determining the ABO blood group, shows tier 1 genome-wide significance, where both genes are associated with VTE [111,112]. *F2* gene encoding for prothrombin and *TSPAN15*, encoding the cell surface transporter protein tetraspanin 15, which regulates cell development, activation, growth, and motility, show tier 2 genome-wide significance, and both also known to be associated with VTE [111,112,113]. *TAP2*, which encodes an ATP-binding cassette transporter, is involved in antigen processing and immunity, and shows tier 2 genome-wide significance. *TAP2* is not known to be associated with VTE or PE [111,114]. There are also associations of *F11* encoding coagulation factor XI, EDEM2 being associated with prothrombin time, and *SLC44A2* being involved in platelet activation, thrombosis, and VTE [111,112,115,116]. Although there are similarities with VTE, there is no significant evidence of a shared genome-wide architecture with IPAH [111]. 

## 8. Does Current Understanding of Pathobiology Translate to CTEPH Management?

Managing CTEPH in the multimodal treatment era with PEA, BPA, and pulmonary hypertension medications have been a success story in improving patients’ functional status and survival [40,117,118,119,120,121]. However, mechanical treatment with PEA and BPA only addresses the end outcome of CTEPH by removing obstructions to improve flow. CTEPH patients with microvasculopathy (inoperable CTEPH and recurrent or persistent CTEPH after PEA) are managed with PH medical therapies which primarily act as vasodilators [7,13,122,123]. Currently, only Riociguat has been licensed worldwide, but off-label use of other PH medications occurs frequently [7,13,122,123]. These PH medications only act on pre-capillary arterioles, and therefore, the variability of their effect could be due to complex CTEPH vascular remodeling affecting pre-capillary vessels and additionally post-capillary venules and bronchial collaterals [3,4]. The current understanding of CTEPH pathobiology has not been translated to PH drug therapy for vascular reverse remodeling. 

Lifelong anticoagulation therapy even after successful mechanical treatment remains vital in preventing new thrombi [7,122,123]. However, the decision between the traditional choice of warfarin (a vitamin K antagonist, VKA) and direct oral anticoagulants (DOACs) remains a contended issue. Although prospective trials comparing warfarin and DOACs in CTEPH are unavailable, recent real-world studies have shown that patients on DOACs are twice as likely to have acute or subacute thrombi at the time of PEA and higher recurrent VTE after PEA [124,125]. Thrombophilia risks have not been found to contribute to the CTEPH pathobiology, but there are data indicating a more central role of vWF, including interaction with the inflammatory pathway [29,30,31,32,33]. Perhaps with better understanding of the pathobiology in CTEPH, we may be able to make better decisions on the choice of anticoagulation.

## 9. Conclusions

CTEPH stems from the non-resolution of thrombi due to multifaceted pathological mechanisms including prothrombosis, fibrinolysis resistance, defective angiogenesis, and inflammation (Figure 2). These pathogenic pathways have an apparent genetic background. However, many of the proposed pathological mechanisms are not unique to CTEPH and may overlap with acute PE/VTE, and therefore do not fully answer the question as to why only a minority of patients develop CTEPH after acute PE. There are perceptible interactions between the proposed pathobiological pathways, especially in the coagulation–inflammation–angiogenesis axis; therefore, it is unlikely that a single isolated mechanism is responsible for the development of CTEPH. The microbiome, infection, circulating microparticles, and the plasma metabolome may also play a role in perpetuating CTEPH. 

Research studies into the etiopathology of CTEPH have been challenging, but genetic and epigenetic studies appear to be the frontier of future research in CTEPH. This will bridge gaps in our knowledge related to the natural history of acute PE to chronic thrombi and CTEPH. Subsequently, we may also better understand how to monitor patients after acute PE with risk factors for CTEPH and possibly how to prevent its development. Furthermore, there are important therapeutic implications with better delineation of the pathobiology in CTEPH.

## Figures and Tables

**Figure 1 biomedicines-12-00046-f001:**
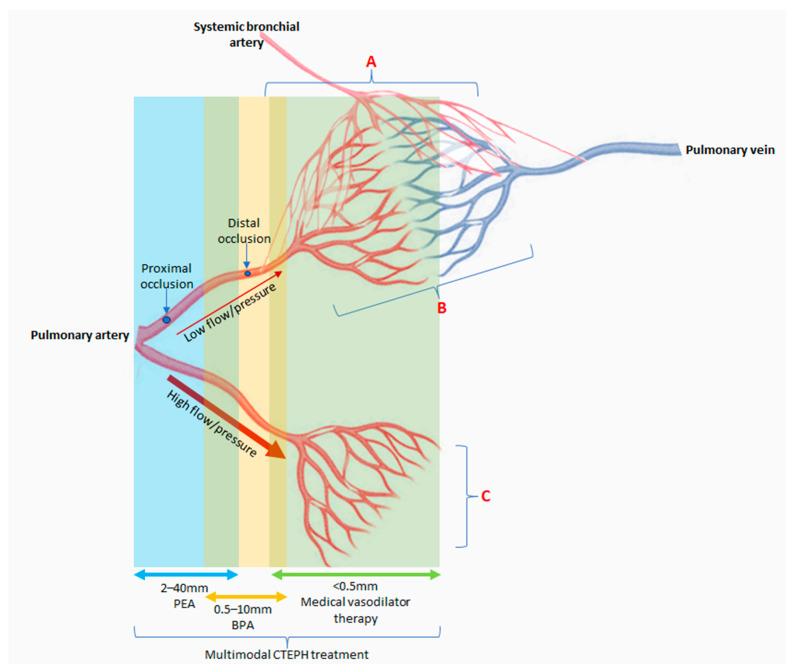
Compartments for multimodal chronic thromboembolic pulmonary hypertension (CTEPH) management and differential microvasculopathy (vessel diameter not to scale). (**A**) Systemic to pulmonary circulation anastomoses which become functional when distal to pulmonary artery occlusions. (**B**) Microvasculopathy distal to obstructed pulmonary arteries are possibly due to anastomoses from systemic bronchial arteries with pre-capillary arterioles, capillaries, and pulmonary venules. (**C**) Microvasculopathy distal to non-obstructed pulmonary arteries are due to high flow/high pressure, resulting in shear stress leading to vascular remodeling. PEA: pulmonary endarterectomy; BPA: balloon pulmonary angioplasty.

**Figure 2 biomedicines-12-00046-f002:**
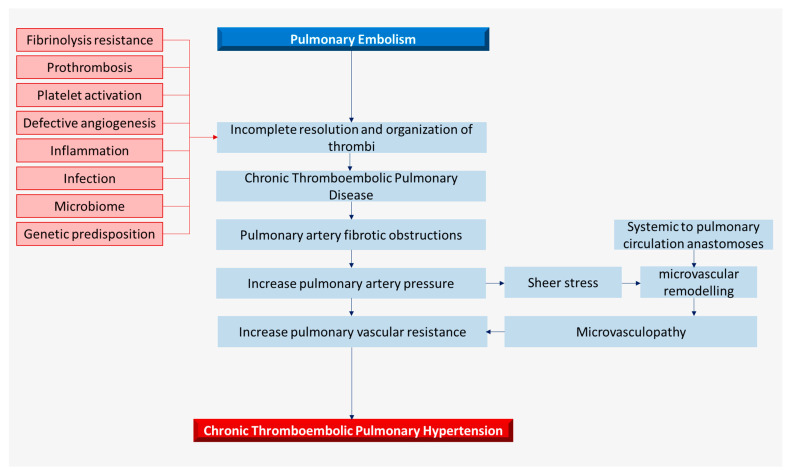
Pathobiology and pathophysiology of CTEPH.

## Data Availability

Not applicable.

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
