# Peer review of "Chronic Thromboembolic Pulmonary Hypertension: A Review of the Multifaceted Pathobiology"

_biomedicines, 2023, doi:10.3390/biomedicines12010046_

Round 1
Reviewer 1 Report
Comments and Suggestions for Authors
The topic is interesting and the paper is quite well written. Nevertheless, in my opinion, some parts need to be improved, I have some comments:
1) In this article, we review the most relevant multifaceted cross-talking 21 pathogenic mechanisms and advances in understanding the pathobiology in CTEPH, as well as its 22 challenges and future direction. Abstract might be beneficial to include a sentence in the abstract that briefly summarizes the key findings of the study. This can provide readers with a quick overview of the research.
2) 1. Introduction 27 Chronic thromboembolic pulmonary disease (CTEPD) is characterised by chronic or- 28 ganized thromboembolic material in the pulmonary arterial tree (1–4). The non-resolving, 29 organized thrombi resulting in pulmonary artery (PA) obstructions and additional micro- 30 vascular remodelling may lead to the development of pulmonary hypertension, and this 31 condition is referred to as chronic thromboembolic pulmonary hypertension (CTEPH) (1– 32 6). Risk factors for CTEPH have been well documented, however how they biologically 33 influence the development of chronic thrombi are poorly understood (1,3,4,7,8). About 34 75% of patients with CTEPH have a history of acute pulmonary embolism (PE) and alt- 35 hough residual thrombi or perfusion defects after acute PE are not uncommon, the rela- 36 tionship between acute PE and progression to chronic thrombi and CTEPH is difficult to 37 elucidate (3,8–11). CTEPH patients without history of acute PE or deep vein thrombosis 38 (DVT) may have sub clinical thromboembolic event. The natural history from acute PE to 39 CTEPD/CTEPH is not fully described or understood. I suggest that you include some information in order to complete the manuscript. Below you can find some works that could give useful ideas in expanding this part, I suggest these references: a- Chronic Thromboembolic Pulmonary Hypertension: An Update. Diagnostics (Basel). 2022 Jan 19;12(2):235. doi: 10.3390/diagnostics12020235; b- Correlation between CT Value on Lung Subtraction CT and Radioactive Count on Perfusion Lung Single Photon Emission CT in Chronic Thromboembolic Pulmonary Hypertension. Diagnostics (Basel). 2022 Nov 21;12(11):2895. doi: 10.3390/diagnostics12112895; c- Chronic Thromboembolic Pulmonary Hypertension: An Observational Study. Medicina (Kaunas). 2022;58(8):1094. doi: 10.3390/medicina58081094. d- Description, Staging and Quantification of Pulmonary Artery Angiophagy in a Large Animal Model of Chronic Thromboembolic Pulmonary Hypertension. Biomedicines 2020, 8, 493. https://doi.org/10.3390/biomedicines8110493
3) Numerous pathobiological mechanisms have been proposed in published medical 58 literature which may be contrasting. Due to the complex nature in the development of 59 CTEPH, multifaceted pathological pathways have been explored to comprehend the bio- 60 logical process. In this article we review the most relevant research findings and discuss 61 the advances in understanding of pathobiology in CTEPH. Please improve the description of this part and underline the novelty of the study.
4) 8. Conclusions 351 Managing CTEPH in the multimodal treatment era with PEA, BPA and pulmonary 352 hypertension medications have been a success story in improving patients’ functional sta- 353 tus and survival (35,106–109). However, there is still limited understanding of the patho- 354 biology. CTEPH stem from non-resolution of thrombi due to multifaceted pathological 355 mechanisms including pro-thrombosis, fibrinolysis resistance, defective angiogenesis, 356 and inflammation (Figure 2). These pathogenic pathways have an apparent genetic back- 357 ground. There are perceptible interactions between the proposed pathobiological .. The conclusion section needs to be improved. It could be interesting to record the aim of the study. It is necessary to be more concise in the presentation of the facts, clarifying the results obtained and comparing them with previous or similar studies. However, it is interesting to answer the questions that arise from these results, backed up by published literature.
Comments on the Quality of English Language
Minor changes of English language are required
Author Response
|
Dear reviewer,
|
|
Response to Reviewer 1 Comments |
|
1. Summary
|
|
Thank you very much for taking the time to review this manuscript. Please find the detailed responses below and the corresponding revisions/corrections highlighted/in track changes in the re-submitted files.
|
|
2. Point-by-point response to Comments and Suggestions for Authors
|
|
Comments 1: In this article, we review the most relevant multifaceted cross-talking pathogenic mechanisms and advances in understanding the pathobiology in CTEPH, as well as its challenges and future direction. Abstract might be beneficial to include a sentence in the abstract that briefly summarizes the key findings of the study. This can provide readers with a quick overview of the research.
|
|
Response 1: Thank you for pointing this out. We agree and for clarity we have now moved the sentence “In this article, we review the most relevant multifaceted cross-talking pathogenic mechanisms and advances in understanding the pathobiology in CTEPH, as well as its challenges and future direction” forwards and further in the text we describe findings from literature review.
These changes are tracked in the Abstract section of the manuscript, line 15 to 17 and line 23 to 25.
|
|
Comments 2: Introduction Chronic thromboembolic pulmonary disease (CTEPD) is characterized by chronic organized thromboembolic material in the pulmonary arterial tree (1–4). The non-resolving, organized thrombi resulting in pulmonary artery (PA) obstructions and additional microvascular remodeling may lead to the development of pulmonary hypertension, and this condition is referred to as chronic thromboembolic pulmonary hypertension (CTEPH) (1–6). Risk factors for CTEPH have been well documented, however how they biologically influence the development of chronic thrombi are poorly understood (1,3,4,7,8). About 75% of patients with CTEPH have a history of acute pulmonary embolism (PE) and although residual thrombi or perfusion defects after acute PE are not uncommon, the relationship between acute PE and progression to chronic thrombi and CTEPH is difficult to elucidate (3,8–11). CTEPH patients without history of acute PE or deep vein thrombosis (DVT) may have sub clinical thromboembolic event. The natural history from acute PE to CTEPD/CTEPH is not fully described or understood. I suggest that you include some information in order to complete the manuscript. Below you can find some works that could give useful ideas in expanding this part, I suggest these references: a- Chronic Thromboembolic Pulmonary Hypertension: An Update. Diagnostics (Basel). 2022 Jan 19;12(2):235. doi: 10.3390/diagnostics12020235; b- Correlation between CT Value on Lung Subtraction CT and Radioactive Count on Perfusion Lung Single Photon Emission CT in Chronic Thromboembolic Pulmonary Hypertension. Diagnostics (Basel). 2022 Nov 21;12(11):2895. doi: 10.3390/diagnostics12112895; c- Chronic Thromboembolic Pulmonary Hypertension: An Observational Study. Medicina (Kaunas). 2022;58(8):1094. doi: 10.3390/medicina58081094. d- Description, Staging and Quantification of Pulmonary Artery Angiophagy in a Large Animal Model of Chronic Thromboembolic Pulmonary Hypertension. Biomedicines 2020, 8, 493. https://doi.org/10.3390/biomedicines8110493
|
|
Response 2: Thank you for the suggested references.
We have expanded paragraph regarding the incidence of CTEPH, including adding the suggested reference “Chronic Thromboembolic Pulmonary Hypertension: An Update. Diagnostics (Basel). 2022 Jan 19;12(2):235. doi: 10.3390/diagnostics12020235” to explain to readers that there is still the remaining question as to why only a minority of patient with acute PE develop CTEPH. These changes are tracked in the Introduction section of the manuscript, line 41 to 47. The added suggested reference is now reference number 13.
We have added the suggested reference “Chronic Thromboembolic Pulmonary Hypertension: An Observational Study. Medicina (Kaunas). 2022;58(8):1094. doi: 10.3390/medicina58081094” to our Conclusion section in the sentence “Managing CTEPH in the multimodal treatment era with PEA, BPA and pulmonary hypertension medications have been a success story in improving patients’ functional status and survival.” This is now reference number 121.
We have added the suggested reference “Description, Staging and Quantification of Pulmonary Artery Angiophagy in a Large Animal Model of Chronic Thromboembolic Pulmonary Hypertension. Biomedicines 2020, 8, 493. https://doi.org/10.3390/biomedicines8110493” in the third paragraph of the introduction section as we agree it will be interesting for readers to read further and we included that animal models may describe possible pathobiological pathways contributing to CTEPH and vascular remodeling. We have described animal models in the relevant sections. These added changes are tracked in the Introduction section of the manuscript, line 61 to 62. The added suggested reference is now reference number 19.
|
|
Comments 3: Numerous pathobiological mechanisms have been proposed in published medical literature which may be contrasting. Due to the complex nature in the development of CTEPH, multifaceted pathological pathways have been explored to comprehend the biological process. In this article we review the most relevant research findings and discuss the advances in understanding of pathobiology in CTEPH. Please improve the description of this part and underline the novelty of the study.
|
|
Response 3: We agree with the reviewer and have expanded this last paragraph of the Introduction section. We expanded on the challenges of previous proposed pathological pathways in CTEPH and limitations of previous studies. We have also added that our article is not only a review of the literature but also makes comparison with diseases that has similar pathological pathways with CTEPH as we find this makes interpretation of proposed mechanisms more challenging.
These changes are tracked in the Introduction section of the manuscript, line 67 to 71 and 74 to 76.
|
|
Comments 4: Conclusions Managing CTEPH in the multimodal treatment era with PEA, BPA and pulmonary hypertension medications have been a success story in improving patients’ functional status and survival (35,106–109). However, there is still limited understanding of the pathobiology. CTEPH stem from non-resolution of thrombi due to multifaceted pathological mechanisms including pro-thrombosis, fibrinolysis resistance, defective angiogenesis, and inflammation (Figure 2). These pathogenic pathways have an apparent genetic background. There are perceptible interactions between the proposed pathobiological... The conclusion section needs to be improved. It could be interesting to record the aim of the study. It is necessary to be more concise in the presentation of the facts, clarifying the results obtained and comparing them with previous or similar studies. However, it is interesting to answer the questions that arise from these results, backed up by published literature.
|
|
Response 4: Thank you for this comment. We have made the Conclusion section more concise to illustrate the relevant pathways and challenges with explaining the pathobiology in CTEPH. We have also included the suggested reference in response to Comment 2.
These changes are tracked in the Conclusion section of the manuscript, line 417 to 426 and 429 to 435.
|
|
3. Response to Comments on the Quality of English Language
|
|
Point 1: Minor changes of English language are required.
|
|
Response: Thank you for this comment. As only minor English changes are required this should be covered by the APC (as per the information for authors MDPI webpage). |
Sincerely,
Dr Hakim Ghani
Clinical Research Fellow
National Pulmonary Hypertension Centre, Royal Papworth Hospital, Cambridge, UK
Supervised and co-authored by
Dr Joanna Pepke-Zaba
Consultant Respiratory Physician
National Pulmonary Hypertension Centre, Royal Papworth Hospital, Cambridge, UK
Affiliated Associate Professor
University of Cambridge, Cambridge

Reviewer 2 Report
Comments and Suggestions for Authors
The authors submitted a narrative review in which they reported the most relevant research findings and discuss the advances in understanding of pathobiology in chronic thromboembolic pulmonary disease. They throughly described pathogenic pathways of the condition and plausible causes turning acute PE to chronic thrombi and CTEPH. The manuscript has a logical structure, and it is well referenced. The figures are legible and clear. The cocnlusive part is informative abd readable. Although the findings of the article seem to be intriguing, I would like to make some comments to discuss.
1. The authors did not report any aspect of a link between inflammation, thrombosis, infection with NETosis, while this issues is considered to be improtant for the pathobiology of the condition. Please, check and give extensive report.
2. Epigenetic regulation of thrombophilia and spontaneous thrombosis seem not to be throughly reported. Please, add this information in the text of the paper.
3. Please, add a brief comment regarding plausible link of novel aspect of pathobiology of CTEPH and a variablity of the therapy responce.
Author Response
|
Dear reviewer,
|
|
Response to Reviewer 2 Comments |
|
1. Summary
|
|
Thank you very much for taking the time to review this manuscript. Please find the detailed responses below and the corresponding revisions/corrections highlighted/in track changes in the re-submitted files.
|
|
2. Point-by-point response to Comments and Suggestions for Authors
|
|
Comments 1: The authors did not report any aspect of a link between inflammation, thrombosis, infection with NETosis, while this issue is considered to be important for the pathobiology of the condition. Please, check and give extensive report.
|
|
Response 1: Thank you for pointing this out. We agree that this was an omission on our side. We have therefore added a paragraph on NETosis in the Metabolomics- Angiogenesis and Inflammation section (4th paragraph): Neutrophil extracellular traps (NETs) formation, NETosis, which is proinflammatory, prothrombotic and proangiogenic is elevated in both plasma and PEA specimens in CTEPH (55–57). Similar findings were also found in idiopathic PAH (IPAH) patients. NETosis resulting from at least three possible mechanisms: cell lytic, vesicle-mediated, and mitochondrial release, of neutrophils result in a net of chromatin fibers with neutrophil secretory granules containing elastase and myeloperoxidase which can induce nuclear factor (NF)-κB and transforming growth factor (TGF) β in CTEPH (55,56,58). Interestingly, a small study showed that NETs form predominantly in the organizing stage of thrombi development which require further research as a possible important mechanism in CTEPH (59). Although NETs support chronic thrombi development by thrombofibrosis and promoting thrombo-inflammation, NETosis seem to be proangiogenic which is counterintuitive to the defective angiogenesis narrative in CTEPH (55,56). However, this may explain the variability of revascularization seen in chronic thrombi. Degradation of NETs with DNases may prevent chronic clot formation when used in the acute phase (55,56,58). These changes are tracked in the Metabolomics- Angiogenesis and Inflammation section of the manuscript, line 196 to 209.
We have also added a sentence in the 1st paragraph of the Microbiome and Infection section stating a study which showed the role of NETs in an IVC ligation mouse model. These changes are tracked in the Microbiome and Infection section of the manuscript, line 245 to 247.
|
|
Comments 2: Epigenetic regulation of thrombophilia and spontaneous thrombosis seem not to be thoroughly reported. Please, add this information in the text of the paper.
|
|
Response 2: Thank you for this suggestion. On further literature review we have now added the role of epigenetic regulation of vWF (which we already note plays a vital pathological role) and in situ thrombosis in CTEPH in the 2nd paragraph of the Genetic Background section: The inflammation-coagulation axis in CTEPH is supported with evidence of epigenetic regulation of the vWF promoter in CTEPH endothelium. Reduced histone tri-methylation and increased histone acetylation of the vWF promoter facilitate binding of NF-κB2 and drive vWF transcription, which in turn increase platelet adhesion (33). This could suggest the epigenetic role in linking inflammation and coagulation pathways to create an environment favoring in situ thrombosis. Although this study compared CTEPH with healthy controls and IPAH, lacked an acute PE/VTE cohort which could give insight whether the epigenetic regulation of vWF is unique to the pathogenesis in CTEPH. These changes are tracked in the Genetic Background section of the manuscript, line 315 to 323.
We had also suggested that the ABO locus is pleiotropic (with over representation of non-O blood group in CTEPH) and could influence genetic regulation in a yet to be discovered way. To emphasize the possible role of epigenetics in CTEPH we have added that further research into chromosome conformation is required as the last sentence in the 3rd paragraph of the Genetic Background section. These changes are tracked in the Genetic Background section of the manuscript, line 338 to 340.
|
|
Comments 3: Please, add a brief comment regarding plausible link of novel aspect of pathobiology of CTEPH and a variability of the therapy response.
|
|
Response 3: Thank you for this valid comment. Apart from pharmacokinetic and pharmacodynamic studies in Riociguat in CTEPH there is limited research in variability of PH medication response in CTEPH. We therefore expanded on the possible therapeutic implication of PH medical therapy with better understanding of CTEPH pathobiology and made a new section titled “Does Current Understanding of Pathobiology Translate to CTEPH Management?”: Managing CTEPH in the multimodal treatment era with PEA, BPA and pulmonary hypertension medications have been a success story in improving patients’ functional status and survival (40,117–121). However, mechanical treatment with PEA and BPA only addresses the end outcome of CTEPH by removing obstructions to improve flow. CTEPH patients with microvasculopathy, (inoperable, and re-current or persistent PH after PEA) are managed with PH medical therapies which primarily act as vasodilators (7,13,122,123). Currently only Riociguat has been licensed worldwide but off-label use of other PH medications occurs frequently (7,13,122,123). These PH medications only act on pre-capillary arterioles and therefore the variability of their effect could be due to complex CTEPH vascular remodeling affecting pre-capillaries vessels and additionally post-capillary venules and bronchial collaterals (3,4). The current understanding of CTEPH pathobiology has not been translated to PH drug therapy for vascular reverse remodeling. Lifelong anticoagulation therapy even after successful mechanical treatment remains vital in preventing new thrombi (7,122,123). However, the decision between the traditional choice of warfarin (a vitamin K antagonist, VKA) and direct oral anticoagulants (DOACs), remains a contended issue. Although prospective trials comparing warfarin and DOACs in CTEPH are unavailable, recent real-world studies have shown that patients on DOACs are twice as likely to have acute or subacute thrombi at the time of PEA and higher recurrent VTE after PEA (124,125). Thrombophilia risks have not been found to contribute to the CTEPH pathobiology but there are data indicating a more central role of vWF including interaction with the inflammatory pathway (29–33). Perhaps with better understanding of the pathobiology in CTEPH, we may be able to make better decisions on the choice of anticoagulation.
These changes are tracked in the new section “Does Current Understanding of Pathobiology Translate to CTEPH Management?”, line 392 to 415.
|
Sincerely,
Dr Hakim Ghani
Clinical Research Fellow
National Pulmonary Hypertension Centre, Royal Papworth Hospital, Cambridge, UK
Supervised and co-authored by
Dr Joanna Pepke-Zaba
Consultant Respiratory Physician
National Pulmonary Hypertension Centre, Royal Papworth Hospital, Cambridge, UK
Affiliated Associate Professor
University of Cambridge, Cambridge, UK

Round 2
Reviewer 1 Report
Comments and Suggestions for Authors
The manuscript has been improved as requested, I have no further comments
Comments on the Quality of English LanguageMinor revisions of English language required